# Vitamin B1 Deficiency and Perimyocarditis Fulminans: A Case Study of Shoshin Syndrome in a Woman Following an Unbalanced Dietary Pattern Followed by a Literature Review

**DOI:** 10.3390/life13010205

**Published:** 2023-01-10

**Authors:** Justyna Rohun, Karolina Dorniak, Krzysztof Młodziński, Witold Bachorski, Marcin Gruchała, Grzegorz Raczak, Ludmiła Daniłowicz-Szymanowicz

**Affiliations:** 1Department of Cardiology and Electrotherapy, Medical University of Gdansk, 80-214 Gdansk, Poland; 2Department of Noninvasive Cardiac Diagnostics, Medical University of Gdansk, 80-214 Gdansk, Poland; 3First Department of Cardiology, Medical University of Gdansk, 80-214 Gdansk, Poland

**Keywords:** myocarditis, beriberi, Shoshin syndrome, vitamin B1 deficiency

## Abstract

(1) Background: vitamin B1 level depletion, known as a beriberi syndrome, can lead to severe cardiovascular complications, from which perimyocarditis fulminans is one of the most severe. (2) Methods: this is a retrospective case study that includes an adult patient with clinical presentations of acute heart failure (HF) symptoms following perimyocarditis on the grounds of thiamine deficiency. (3) Results: A 49-year-old woman presented with acute HF symptoms due to perimyocarditis. The patient suddenly developed refractory cardiogenic shock with metabolic acidosis requiring maximal medical management, including an intra-aortic balloon pump and extracorporeal membrane oxygenation. Due to additional peripheral polyneuropathy, beriberi disease was suspected after excluding other possible causes of the patient’s condition. After administration of vitamin B1, clinical improvement in the patient’s condition and the resolution of metabolic abnormalities were observed, which ultimately confirmed the diagnosis of Shoshin syndrome caused by the implementation of a gluten-free diet without indications for its adherence. (4) Conclusions: Fulminant beriberi disease, although considered rare, is a life-threatening condition and should always be included in the differential diagnosis of critically ill patients, notably those with malnutrition. An unbalanced diet can be detrimental and have severe consequences, i.e., perimyocarditis fulminans. However, treatment with thiamine can significantly improve the patient’s cardiac function and restore hemodynamic and metabolic parameters.

## 1. Introduction

Thiamine, or aneurine, a small, water-soluble vitamin in the B complex family, plays a crucial role in many cellular processes [1,2]. In its activated form, thiamine pyrophosphate (TPP) acts as an enzymatic cofactor for several reactions during carbohydrate metabolism. The most important aneurine-dependent biocatalysts are pyruvate dehydrogenase (PDH) and 2-oxoglutarate dehydrogenase (OGDH), which enable pyruvate, a product of glycolysis, to enter the citric acid cycle, and, therewith, promote aerobic metabolism and adenosine triphosphate (ATP) production [1,2]. The reactions mentioned above are known as the central metabolic pathway which provides cells with their energetic requirements. Consequently, the importance of thiamine is predominantly noticed in organs with the highest metabolic activity, including those in the cardiovascular, nervous, and musculoskeletal systems [1,2,3]. 

A depletion in vitamin B1 levels is called beriberi syndrome. The two main manifestations of this disease affect either the neuromuscular system (a dry form of beriberi) or the cardiovascular system (a wet form of beriberi) [4]. Rarely, a fulminant version called Shoshin syndrome may be seen [5,6,7,8,9]. It is characterized by low cardiac output and hemodynamic collapse. Severe cardiac dysfunction and lactic acidosis are observed [6,7,8,9]. The course of this form of the disease can be fulminant, and when untreated, it can be fatal [10]. Due to complex clinical presentation, non-specific symptoms, and a lack of clinically approved diagnostic tests, Shoshin syndrome can be easily misdiagnosed [10]. 

The current paper aims to present the case of a 49-year-old woman who suddenly developed perimyocarditis fulminans complicated by cardiogenic shock due to thiamine deficiency. Difficulties in the differential diagnosis, the importance of a thorough medical history, and the effect of optimal medical treatment, including mechanical circulation support and intravenous vitamin B1 administration are described. We highlight the practical implications of the study, and, finally, a summary review of the available literature on Shoshin syndrome is provided.

## 2. Materials and Methods

This is a single-center, observational case study of fulminant perimyocarditis during beriberi syndrome. We retrospectively identified patients between the periods of January 2016 and December 2020, who were admitted to the Clinical Centre of Cardiology of the University Clinical Centre in Gdańsk, Poland, with the symptoms of new-onset heart failure (HF) and had the features of perimyocarditis resulting from vitamin B1 deficiency confirmed either by laboratory tests or by showing clinical improvement after thiamine administration. Additionally, other distinguishable causes of perimyocarditis among patients were excluded. Patients with normal thiamine levels were not included. Physical examination, laboratory data, echocardiography results, cardiac magnetic resonance imaging, and endomyocardial biopsy results were taken into consideration for further analysis. As part of the literature review, the PubMed database was searched for the literature, clinical trials, and databases from 1978 to December 2022. The keywords were “Shoshin syndrome” or “fulminant beriberi”. Case reports were mostly found.

## 3. Case Presentation

A 49-year-old female with a short history of previous de novo HF based on perimyocarditis requiring hospitalization (Figure 1) was re-hospitalized due to HF exacerbation. 

At the time of admission, the patient was hemodynamically stable but complained of dyspnea, nausea, and weakness. On physical examination, mild lower limb edema and pale skin color were noticeable. In a 12-lead electrocardiogram, a sinus rhythm of 110 beats per minute (BPM) was observed with unspecific ST-T segment changes (Figure 2). 

The patient’s condition suddenly deteriorated to cardiogenic shock within a few hours. Pronounced lactic acidosis with a pH of 6.6 and lactate level of 19 mmol/L was identified (Table 1). 

Myocardial injury markers were slightly elevated in the laboratory assays. Troponin I (TnI) was 1.01 ng/mL, creatinine kinase-MB mass (CK-MB mass) was 27.6 ng/mL), and brain natriuretic peptide (BNP) was significantly increased—up to 1800 pg/mL (Table 1). A transthoracic echocardiography (TTE) revealed severe biventricular dilatation and failure (left ventricular ejection fraction (LVEF) of 10% without regional wall motion disturbances and tricuspid annular plane systolic excursion of 8 mm). Furthermore, moderate tricuspid and mitral regurgitations and a hemodynamically insignificant amount of pericardial fluid with dilated non-collapsible with inspiration inferior vena cava were observed. Due to progressive respiratory failure, the patient was intubated, and mechanical ventilation was initiated. Despite intensive pharmacological treatment (including fluid administration and high doses of vasopressors, such as adrenaline, noradrenaline, and dobutamine), no increase in blood pressure was obtained, with maximum values reaching 50/30 mmHg. An intra-aortic balloon pump (IABP) was implanted, and further, due to its inefficacy, an emergency venoarterial extracorporeal membrane oxygenation (VA-ECMO) supply was implemented. The procedure was complicated by bleeding into the soft tissue of the right lower limb (with the hemoglobin level decreasing to 6.4 g/dL); therefore, repeated blood transfusions were performed with a slight consequent increase in the hemoglobin level rising up to 8.4 g/dL (Table 2). 

Repeated CXR examinations revealed persistent, severe pulmonary edema, notably in the hilar regions of both lungs, despite maintaining negative fluid balance. 

In the meantime, there was an incidence of flaccid paraparesis observed, notably involving both feet. Neurological examination revealed symmetrical impairment of motor, sensory, and reflex functions of lower limbs without any neurological deficit in the upper limbs, so the neuropathy–myopathy of the critical state was suspected, and supplementation of coenzyme Q10, selenium, and D3 vitamin was started along with parenteral nutrition. 

In the following days of hospitalization, the features of acute kidney injury were observed: creatinine was 3.37 mg/dL, blood urea nitrogen (BUN) was 101.1 mg/dL, and the estimated glomerular filtration rate (eGFR) was 15 mL/min/1.73 m^2^ (Table 1), and despite diuresis being present, continuous renal replacement therapy with continuous venovenous hemofiltration (CRRT—CVVHF) was initiated. Due to later ventilator-associated pneumonia, targeted antibiotic therapy (with meropenem) was used with sound clinical effects. Various laboratory data derived from the patient are presented in Table 2. 

With time, significant improvement in clinical status was observed (Table 1) with gradual improvement in LVEF (eventually up to 60%); therefore, CRRT—CVVHF and VA-ECMO were decannulated, and subsequently, the patient was successfully extubated. For optimal HF treatment, an angiotensin receptor blocker, beta blocker, mineralocorticoid receptor antagonist, and loop diuretic were continued. Moreover, due to neurological deficits, tailored physical rehabilitation was provided along with an easily digested balanced diet and supplementation of polyunsaturated fatty acids, selenium, and vitamins. 

Further, an extensive panel of additional studies was made to determine the leading cause of perimyocarditis fulminans (Table 3). 

Despite a wide range of diagnostic tests, an unambiguous cause of disease could not be determined. An electroneurography assay, requested due to persistent flaccid paraparesis, revealed marked, symmetrical, axonal demyelinating polyneuropathy of both lower limbs, suggesting beriberi disease. The diagnosis was made based on coexisting peripheral neuropathy and cardiovascular failure with severe lactate acidosis. Additionally, due to the thorough medical history obtained, the patient was following a long-lasting imbalanced gluten-free diet, adopted without medical consultation, due to an unconfirmed celiac disease. Even though the thiamine blood level was not obtained in the presented case, based on the specific clinical picture, intramuscular thiamine replacement therapy was empirically initiated (50 mg once daily) with an adequate clinical response. The patient’s cardiovascular and renal functions quickly and markedly improved (LVEF raised to 70%, creatinine was 0.73 mg/dL, and eGFR was >90 mL/min/1.73 m^2^) with lower limb polyneuropathic symptoms markedly mitigated.

After a nutritional consultation, the patient was discharged home in good general condition. The lady was informed about the negative consequences of an unbalanced gluten-free diet, which is harmful in people without confirmed celiac disease. Within a year, the lady saw an improvement in exercise tolerance, allowing her to return to physical activity.

## 4. Discussion

The main finding of our study reveals that vitamin B1 deficiency can be a life-threatening condition, and therapy with thiamine results in a significantly favorable clinical response. 

In the absence of thiamine, pyruvate cannot be an input for a Krebs cycle and is converted into lactate following an anaerobic process [1,2]. A less efficient anaerobic respiration resulting in ATP depletion does not meet energetic organ demand and leads to a decrease in function, i.e., myocardial contraction, which, over time, can manifest as cardiogenic shock, as seen in our patient. Moreover, with lactate accumulation, metabolic acidosis usually presents, which we noticed during the hospitalization of our lady. A long-term decrease in the thiamine level caused by its insufficient consumption by following an unbalanced dietary pattern eventually led to decompensation of the cardiovascular system manifesting itself as refractory circulatory collapse, resistance to standard HF therapy, and the need for medical management, even with mechanical circulation support. Subsequently, acute HF caused multiorgan failure (particularly noted within the renal, gastrointestinal, and neural systems—Table 1) which significantly complicated the hospitalization course and was potentially fatal.

Currently, Shoshin syndrome is considered rare in developed countries. However, its exact incidence remains unknown [11]. The most common causes include excessive alcohol consumption, liver disease and malnutrition [1,2,4,8,9,10]. Therefore, an unbalanced diet, including a gluten-free diet, which has become increasingly popular in recent decades, should be seriously considered in the diagnostic algorithm for this pathology as demonstrated in the presented patient. Since the human body cannot synthesize thiamine, it requires dietary supplementation. The most common nutritional sources are whole grains, bread, and pulses, while fruits and vegetables, the fundamentals of the gluten-free regimen, are less valuable [1,2]. Thus, a human’s dietary habits are of great importance and can be remarkably detrimental if left unbalanced, as observed in our patient [12]. The patient had adopted a gluten-free diet, depleted of micronutrients, including vitamin B1, without any medical consultation or medically proven gluten-related disease, which resulted in cardiogenic shock on the grounds of perimyocarditis in the course of fulminant beriberi disease. Nowadays, it is considered that gluten could be harmful for the human body. However, available scientific data deny this hypothesis as it is proven that gluten restriction may have severe adverse effects, as were present in our patient [13,14]. 

Due to low awareness of Shoshin syndrome among physicians, unrevealing symptoms involving multiple systems, and a lack of specific, widely obtainable laboratory assays, the diagnosis of fulminant beriberi disease is extremely challenging [10]. In the literature, many cases of wet beriberi are described [4,5,6,7,8,9,11]. However, the symptoms are non-specific and include, among others, acute right-sided HF, biventricular failure, diffuse ST-segment elevation, or T wave inversion [4,5,6,8,9,15]. Likewise, in our case, none of the symptoms in the initially collected anamnesis (no liver disease or excessive alcohol intake) was suggestive of beriberi disease, which delayed the final diagnosis. On admission, our patient was hemodynamically stable, presented the symptoms of acute biventricular failure, hypotension, and metabolic acidosis of unknown cause in a few hours. 

In the diagnostic process of Shoshin syndrome, laboratory blood tests are beneficial and allow confirmation of the disease with certainty. These tests include red blood cell transketolase activity, measurement of the pyrophosphatase effect, and the plasma and intraerythrocytic thiamine levels [16]. However, all of them are only performed in highly specialized centers, and the measurement is time-consuming, complicated, and thereby, infrequent [17,18]. Since obtaining a result is a lengthy procedure, the tests are clinically inefficient, particularly in life-threatening clinical conditions as presented in our patient. However, available data indicate that while suspecting its deficiency, treatment with vitamin B1 is considered safe and beneficial for the diagnosis and final confirmation of cardiovascular beriberi disease [19]. The therapy is outstandingly efficient in Shoshin syndrome and can quickly improve the patient’s condition even after one dose of thiamine [5,6,7,8,9,19]. Therefore, thiamin administration was initiated in our patient based on the presented clinical picture, notably cardiogenic shock with lactic acidosis and neurological features, which resulted in a pronounced clinical improvement with LVEF restoration and subsidence of peripheral neuropathy and finally confirmed the diagnosis of fulminant beriberi disease. The patient was discharged home in good general condition, and, in a follow-up, it was reported she had returned to regular daily activities, including intensive physical activity. 

## 5. Practical Implications

With our article, we aimed to hopefully raise awareness of beriberi disease among practicing clinicians, expand the knowledge about thiamine relevance, and show potential therapeutic options. Although considered rare, vitamin B1 deficiency should always be included in the differential diagnosis of critically ill patients as the condition can be fatal if left misdiagnosed or untreated. Treatment with vitamin B1 is considered safe and generally accepted in patients suspected of its deficiency. Available data indicate that this method could be useful in the treatment of wet beriberi and is confirmed in our study. Hopefully, our experience will contribute to a further understanding of beriberi disease and the impact of thiamine on human health.

## 6. Review of the Literature

To further support of our findings, we performed an extensive literature search of articles published from 1978 up to December 2022. The reference lists of all the included studies were manually searched for other potential eligible studies. We identified 41 papers, from which the majority were case reports (32 papers) and case series. Two original research studies were found. Data include reports from countries all over the world, and the studied population was aged from pre-term infants to the elderly (87 years old). The background of thiamine deficiency varies; however, the clinical picture is identical. All of the patients developed sudden cardiogenic shock with lactic acidosis, and a pronounced improvement of HF after thiamine administration was observed in all of the cases. In one study, patients died due to non-implementation of the proper (thiamine) therapy [10]. A summary is presented in Table 4.

## 7. Conclusions

Although considered rare in developed countries, fulminant beriberi disease is a life-threatening condition. It should always be included in the differential diagnosis of critically ill patients, notably those with malnutrition. An unbalanced diet, i.e., gluten-free, lacking essential microelements can have severe consequences, including perimyocarditis fulminans, which can be fatal if left untreated. However, targeted therapy with thiamine can significantly improve the patient’s cardiac function and restore hemodynamic and metabolic parameters.

## Figures and Tables

**Figure 1 life-13-00205-f001:**
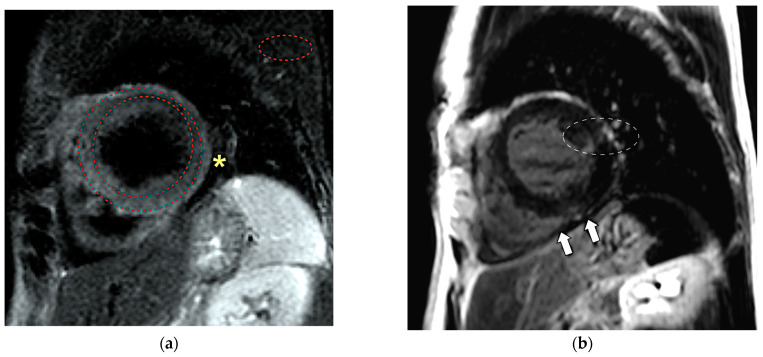
Cardiac magnetic resonance findings in the patient during the previous hospitalization: (**a**) Features of myocardial edema (significant hyperintensity of the myocardium), consistent with ongoing inflammation, can be noted in the short tau inversion recovery T2-weighted (T2STIR) image. The signal intensity (SI) ratio of the myocardium to the skeletal muscle SI is greater than 2, as shown by the red dashed line ROIs. An asterisk marks a small pericardial effusion. (**b**) Subepicardial and intramural streaks of late gadolinium enhancement, pointing out areas of predominantly irreversible damage (arrows) in the inversion recovery images, and possibly related to healed myocarditis. A white dashed line marks an artifact.

**Figure 2 life-13-00205-f002:**
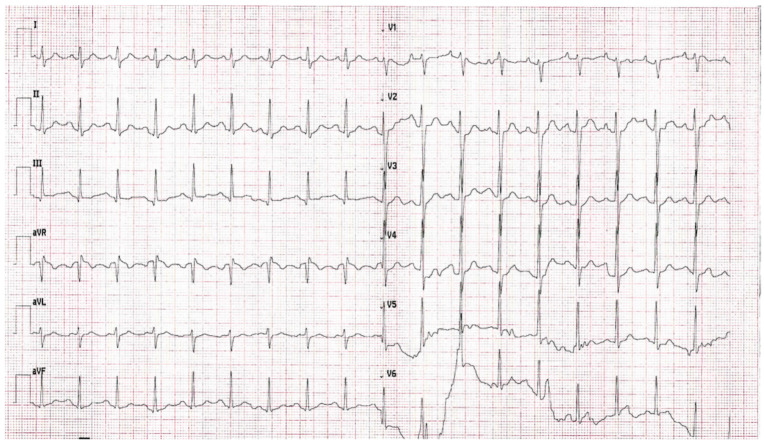
12-lead electrocardiogram of the patient. Sinus tachycardia (110 bpm) with unspecific ST-T segment changes are seen.

**Table 1 life-13-00205-t001:** Time course of basic laboratory assays.

	On Admission	Day 2	Day 5	After 2 Weeks
pH (normal: 7.37–7.45)	6.6	6.9	7.43	7.39
Lac; mmol/L (normal: 0.5–2.2)	19.0	23.0	3.2	1.0
HCO_3_^−^; mmol/L (normal: 22–28)	3.4	7.5	24.4	21.5
4 pCO_2_; mmHg (normal: 31–43)	29.5	41.1	36.4	35.8
BE; mmol/L (normal: −2.0–3.0)	−34.6	−24.4	0.3	−2.7
TnI; ng/mL (normal: 0–0.1)	1.01	0.9	19.25	0.02
CK-MB mass; ng/mL (normal: 0.0–6.6)	27.6	59.9	250.8	3.9
creatinine; mg/dL (normal: 0.55–1.02)	0.82	1.34	3.37	0.73
eGFR; ml/min/1.73 m^2^	84	47	16	>90
CRP; mg/L (normal: 0.0–5.0)	0.92	105.9	21.74	1.11
PCT; ng/mL (normal: 0.0–0.5)	0.08	7.16	3.34	0.07
WBC; ×10^9^/L (normal: 4–10)	9.43	21	12.15	7.48

Abbreviations: BE—base excess; CK-MB mass—creatine kinase-MB; CRP—C-reactive protein; eGFR—estimated glomerular filtration rate; HCO_3_—bicarbonate; ICU—intensive care unit; Lac—lactate; pCO_2_—partial pressure of carbon dioxide; PCT—procalcitonin; TnI—troponin I; WBC—white blood cells.

**Table 2 life-13-00205-t002:** Further laboratory data derived from the patient.

Hb; g/dL (normal: 12–16)	
on admission	12.5
the outermost value	6.4
iron; ug/dL (normal: 50–170)	38
ferritin; ng/mL (normal: 4.63–204)	588
transferrin; G/L (normal: 1.8–3.82)	1.29
transferrin saturation; % (normal: 15–45)	20.9
ALT; U/L (normal: 0–55)	
on admission	31
the outermost value	495
AST; U/L (normal: 0–35)	
on admission	21
the outermost value	555
serum albumin; G/L (normal: 35–50)	18

Abbreviations: ALT—alanine transaminase; APTT—activated partial thromboplastin time; AST—aspartate transaminase; Hb—hemoglobin; INR—international normalized ratio; PT—prothrombin time.

**Table 3 life-13-00205-t003:** Further studies performed to establish the final diagnosis.

Endomyocardial Biopsy	Cardiac Fragments within Normal Limits
anti-beta2GPI antibodies IgG (normal 0–5)	0.27
anti-beta2GPI antibodies IgM (normal 0–5)	0.00
aPS antibodies IgG (normal 0–10)	3.48
aPS antibodies IGM (normal 0–10)	0.00
dRVVT—LA1 (screening); sec (normal 31–44)	27
anti-cardiac muscle antibodies (normal 0–1)	1:10
anti-striated muscle antibodies (normal 0–1)	1:10
ANCA (normal 0–1)	0
HLA B27 antigen	positive
anti-CMV antibodies IgG; U/ml	26.8
anti-CMV antibodies IgM; U/ml	<5
anti-borelia burgdorferi antibodies IgM and IgG	negative
Chlamydophilia pneumoniae IgA, IgM and IgG	negative
Mycoplasma pneumoniae IgA, IgM and IgG	negative
Clostridium difficile toxins (A, B, GDH)	negative
Pneumocystis carinii trophozoits	negative
adenoviruses and rotaviruses	negative
eosinophil count; ×10^9^/L (normal 0.02–0.5)	0.17
eosinophil count; ×10^9^/L (normal 0.02–0.5)	2.3
TSH; uU/mL (normal 0.34–4.94)	5.25
fT4; pmol/L (normal 9.01–19.05)	15.17
fT3; pmol/L (normal 2.63–5.7)	3.49
anti-tTg IgA; RU/mL (normal 0–20)	11.55
anti-tTg IgG	0.05
total IgA; G/L (normal 0.65–4.21)	2.02
total IgM; G/L (normal 0.33–2.93)	14.92
total IgG; G/L (normal 5.52–16.31)	0.96
serum proteins electrophoresis:	
albumin; % (normal 55.8–66.1)	53.2
albumin; G/L (35–50)	28
alpha-1-globulins; % (normal 2.9–4.9)	4.2
alpha-2-globulins; % (normal 7.1–11.8)	7.7
beta-2-globulins; % (normal 3.2–6.5)	6.0
beta-1-globulins; % (normal 4.7–7.2)	6.2
gamma-globulins; % (normal 11.1–18.8)	22.7
total protein G/L (normal 64–83)	66

Abbreviations: ANCA—anti-neutrophil cytoplasmic antibodies; anti-beta2GPI—anti-beta 2 glycoprotein I; anti-tTg—anti-tissue transglutaminase; aPS—anti-antiphosphatidylserine; dRRVT—dilute Russell’s viper venom time; fT3—free triiodothyronine; fT4—free thyroxine; IgA—immunoglobulin A; IgG—immunoglobulin G; IgM—immunoglobulin M; LA—lupus anticoagulant; TSH—thyroid stimulating hormone.

**Table 4 life-13-00205-t004:** Available literature findings of Shoshin beriberi.

Author	Date and Country of the Study	Patient’s Age	Cause of Thiamine Deficiency	Clinical Presentation
Original research				
Barennes, H.; et al. [20]	Laos, 2015	1–6 months old	breastfed infant of thiamine-deficient mother (dietary insufficiency—excessive consumption of polished rice)	acute heart failure (AHF) and lactic acidosis
Bhat J.I., et al. [21]	India, 2017	78.45 ± 30.7 days	breastfed infants of thiamine-deficient mothers (due to customary dietary restrictions)	cardiogenic shock and renal failure
Case series				
Thomas, L.; et al. [22]	France, 1985	3 adult patients, 1 had Shoshin beriberi	excessive alcohol intake	cardiogenic shock and lactic acidosis
Pang, J. A.; et al. [11]	United Kingdom (UK), 1986	2 adult patients	dietary deficiency	AHF with renal failure and severe metabolic acidosis
Shivalkar, B.; et al. [23]	Belgium, 1998	2 adult patients	excessive alcohol intake	one with high-output heart failure (HF) and the other with low-output HF and cardiovascular collapse
Bello, S.; et al. [10]	Italy, 2011	2 patients, 47 and 43 years old	1. gastric resection 2. total parenteral nutrition (TPN)	cardiac arrest and lactic acidosis
Saya, R.P.; et al. [24]	India, 2012	6 patients, 20–36 years old	five were non-alcoholics and none had signs of overt malnutrition	cardiogenic shock, severe metabolic acidosis, and multiorgan failure (MOF)
Dabar, G.; et al. [9]	Lebanon, 2015	4 patients, 23–60 years old	critically ill non-septic non-alcoholic patients 1. gastric surgery 2. pancreatic cancer 3. type-1 glycogen storage disease 4. peritonitis	severe lactic acidosis and refractory cardio-circulatory collapse
Salvatori, G., et al. [25]	Italy, 2016	2 pre-term infants	prolonged TPN	AHF with MOF and lactic acidosis
Case reports				
Attas, M.; et al. [6]	United States of America (USA), 1978	36 years old	excessive alcohol intake	AHF with lactic acidosis and dyspnea
Delorme, N.; et al. [26]	France, 1986	age not precise, adult	excessive alcohol intake	cardiogenic shock with lactic acidosis and severe hyponatremia
Naidoo, D.P.; et al. [27]	Republic of South Africa, 1989	age not precise, adult	TPN and protracted vomiting from intestinal obstruction	refractory cardiogenic shock and lactic acidosis
Cage, J. B.; et al. [28]	USA, 1992	29 years old	end-stage kidney failure on continuous peritoneal dialysis	cardiogenic shock with fulminant pulmonary edema
Debuse, P. J.; et al. [29]	Australia, 1992	3 months old	breastfed infant of thiamine-deficient mother	cardiogenic shock with lactic acidosis
Fujita, I.; et al. [30]	Japan, 1992	21 months old	excessive isotonic sports drink intake	cardiogenic shock with lactic acidosis
Smith, S. W. [31]	USA, 1998	39 years old	excessive alcohol intake	unexplained lactic acidosis and high output HF
López Gastón, O. D.; et al. [32]	Argentina, 2002	58 years old	excessive alcohol intake	dyspnea, oliguria, edema, high output HF, metabolic acidosis, and renal tubular dysfunction
Groeneveld, J.H.; et al. [33]	The Netherlands, 2003	45 years old	excessive alcohol intake, dietary deficiency, and inhalation of salbutamol	cardiogenic shock, severe dyspnea, and lactic acidosis
Kountchev, J.; et al. [8]	Austria, 2005	50 years old	excessive alcohol intake	AHF, dyspnea, and lactic acidosis
Goto, Y.; et al. [34]	Japan, 2007	44 years old	dietary deficiency	cardiogenic shock and pulmonary hypertension
Daly, M.J.; et al. [35]	UK, 2009	post-menopausal woman	prolonged vomiting and diarrhea	cardiogenic shock
Greenspon, J.; et al. [36]	USA, 2010	pediatric patient, age not precise	short bowel syndrome	cardiogenic shock and lactic acidosis
Loma-Osorio, P.; et al. [37]	Spain, 2011	35 years old	not precise	cardiogenic shock and an electrocardiographic pattern of severe myocardial ischemia suggesting left main coronary artery obstruction
Reinhardt, D.; et al. [38]	Switzerland, 2011	67 years old	excessive alcohol intake	acute dyspnea, AHF, and severe lactic acidosis
Brown, T.M.; et al. [39]	USA, 2013	55 years old	excessive alcohol intake and dietary deficiency	AHF and lactic acidosis
Kim, J.; et al. [5]	South Korea, 2014	87 years old	TPN	Cardiogenic shock with diffuse ST-segment elevation (inferior and anterior wall)
Kuno, T.; et al. [40]	Japan, 2014	66 years old	dietary deficiency	AHF, colliquative myocytolysis, and lactic acidosis
Misumida, N.; et al. [41]	Japan, 2014	61 years old	long-term diuretics therapy	AHF
Moulin, P.; et al. [42]	France, 2014	10 weeks old	breastfed infant of thiamine-deficient mother due to high consumption of polished rice	diarrhea, vomiting, lactic acidosis and AHF
Tejedor, A.; et al. [43]	Spain, 2014	45 years old	excessive alcohol intake and dietary deficiency	AHF and dilated cardiomyopathy
Imamura, T.; et al. [44]	Japan, 2015	40 years old	excessive alcohol intake	AHF and lactic acidosis
Cottini, M.; et al. [7]	Italy, 2016	17 years old	not precise	cardiogenic shock and lactic acidosis
Shah, A.; et al. [45]	USA, 2016	33 years old	malnutrition (restrictive, unbalanced diet)	cardiogenic shock and lactic acidosis
Lei, Y.; et al. [19]	China, 2018	39 years old	excessive alcohol intake	AHF, MOF, and lactic acidosis
Elias, I. M.; et al. [46]	Canada, 2019	14 years old	end-stage kidney failure	AHF and lactic acidosis
Vicinanza, A.; et al. [47]	Belgium, 2019	6 years old	Malnutrition and refeeding syndrome	cardiogenic shock with cardiac arrest and severe accidental hypothermia
Didisheim, C.; et al. [48]	Switzerland, 2020	13 years old	TPN	refractory cardiogenic shock and lactic acidosis
Hodgkinson, L. M.; et al. [49]	USA, 2020	68 years old	oral and cutaneous graft-versus-host disease after allogeneic hematopoietic cell transplant (reduced intestinal absorption)	AHF, renal failure, and lactic acidosis
Lim, M. S.; et al. [50]	UK, 2021	57 years old	excessive alcohol intake	cardiogenic shock and lactic acidosis
Tamaki, H.; et al. [51]	Japan, 2022	82 years old	hemodialysis and biguanide therapy	AHF and lactic acidosis
Tanné C.; et al. [52]	Mayotte, 2022	2 months old	breastfed infant of thiamine-deficient mother due to high consumption of polished rice	AHF and lactic acidosis

## Data Availability

Upon reasonable request, data presented in this study will be provided by emailing the corresponding author.

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
