# Peer review of "Vitamin B1 Deficiency and Perimyocarditis Fulminans: A Case Study of Shoshin Syndrome in a Woman Following an Unbalanced Dietary Pattern Followed by a Literature Review"

_life, 2023, doi:10.3390/life13010205_

Round 1

Reviewer 1 Report

"Patient consent was waived by the due to Ethics Committee" ?

Case reports must have a written consent of the patient

Reviewer 2 Report

The authors presented a retrospective case study that includes an adult patient with clinical presentations of acute heart failure (HF) symptoms following perimyocarditis on the grounds of thiamine deficiency.

I have the following concerns:

1. Please provide detailed description of TTE.

2. How can be patients with vitamine B deficiency identified?

3. What are the practical implications of the study?

4. Could you please show the patient's ECG?

Reviewer 3 Report

the case is interesting. I would like the authors to provide a table gathering similar findings in the literature. A sort of review of the literature would improve the paper.

Round 2

Reviewer 1 Report

-

Author Response

Dear Reviewer, we are grateful for your valuable suggestions. The informed consent statement has been sent to the Editors of Life.